# Synthesis of Novel Polymer-Assisted Organic-Inorganic Hybrid Nanoflowers and Their Application in Cascade Biocatalysis

**DOI:** 10.3390/molecules28020839

**Published:** 2023-01-14

**Authors:** Joana F. Braz, Nadya V. Dencheva, Marc Malfois, Zlatan Z. Denchev

**Affiliations:** 1IPC—Institute for Polymers and Composites, University of Minho, 4800-056 Guimarães, Portugal; 2ALBA Synchrotron Facility, Cerdanyola del Valés, 0890 Barcelona, Spain

**Keywords:** hybrid nanoflowers, polyamide microparticles, synchrotron WAXS/SAXS, polymer-assisted biocatalysts, enzyme kinetics, GOx/HRP enzyme cascade

## Abstract

This study reports on the synthesis of novel bienzyme polymer-assisted nanoflower complexes (PANF), their morphological and structural characterization, and their effectiveness as cascade biocatalysts. First, highly porous polyamide 6 microparticles (PA6 MP) are synthesized by activated anionic polymerization in solution. Second, the PA6 MP are used as carriers for hybrid bienzyme assemblies comprising glucose oxidase (GOx) and horseradish peroxidase (HRP). Thus, four PANF complexes with different co-localization and compartmentalization of the two enzymes are prepared. In samples NF GH/PA and NF GH@PA, both enzymes are localized within the same hybrid flowerlike organic-inorganic nanostructures (NF), the difference being in the way the PA6 MP are assembled with NF. In samples NF G/PAiH and NF G@PAiH, only GOx is located in the NF, while HRP is preliminary immobilized on PA6 MP. The morphology and the structure of the four PANF complexes have been studied by microscopy, spectroscopy, and synchrotron X-ray techniques. The catalytic activity of the four PANF was assessed by a two-step cascade reaction of glucose oxidation. The PANF complexes are up to 2–3 times more active than the free GOx/HRP dyad. They also display enhanced kinetic parameters, superior thermal stability in the 40–60 °C range, optimum performance at pH 4–6, and excellent storage stability. All PANF complexes are active for up to 6 consecutive operational cycles.

## 1. Introduction

In the last decades, cascade biocatalysis has attracted significant interest from both academia and industry as a promising technology for green and sustainable chemical production. Unlike the step-by-step synthesis, cascade biocatalysts accomplish two or more consecutive reactions in one-pot, thus avoiding isolation and purification of intermediates. Such procedure has become more efficient due to reduced reaction time and production costs, meanwhile not creating toxic wastes [1]. Nowadays, numerous drugs and fine chemicals [2,3], including chiral alcohols and amines [4,5], carbohydrates [6], and polymers [7] are synthesized through enzyme cascade catalysis. However, the industrial use of cascade biocatalysts based on free enzymes is still problematic, due to their poor stability and difficult reuse [8]. 

A possible way to overcome these limitations is the application of immobilization technologies, in which two or more enzymes are co-localized close to one another on a suitable support. Various inorganic materials e.g., metal oxides, minerals, carbon allotropes, and also organic materials such as synthetic and natural biopolymers have been widely used as supports for enzyme immobilization [9]. Immobilization of enzymes results in a twofold gain: on the one hand, biocatalysts become more resistant to the harmful effects of the environment, e.g., temperature, pH, and solvents; on the other hand, enzymes can be made easily removable from the reaction medium, especially if the carrier applied is with magnetic properties [10]. It has been also demonstrated [1] that in many cases, the co-immobilization boosts the activity of the multienzyme complex. The nature of this effect is still debated, being attributed either to reduction of mass transfer limitations (substrate channeling) [11,12,13] or to a complete change in the microenvironment [14,15,16]. 

Three main methods are widely used for enzyme immobilization, namely (i) binding to a support, (ii) encapsulation or (iii) cross-linking [10]. Each immobilization technique has its advantages and disadvantages. The selection of the method to apply depends on the specific objectives and applications of the target enzyme complex [17,18]. 

The multienzyme cascade complexes can be assembled through: (i) simple fusion of enzymes combining their polypeptide chains by short cross-linkers [19]; (ii) attaching of enzymes on desired docking sites of a scaffold biomolecule, e.g., DNA [20], or (iii) co-immobilization in which the enzymes are bonded to the same support in close physical proximity, mimicking the multienzyme complexes in living cells [21]. The last method is the most frequently used to design cascade catalysts with industrial applications. 

Several approaches have been developed for multienzyme co-immobilization, e.g., random or positional co-immobilization [22,23,24] and compartmentalization [25]. The latter approach has been developed to mimic natural enzyme organization in cells, wherein the enzymes are spatially separated in different patterns and ratios on the same support. Several carriers such as polymer capsules [25], polymersomes [26], liposome vesicles [27], and nano/microsized particles [28] have been used to integrate enzymes via encapsulation or layer-by-layer surface attachment.

Along with their enhanced stability and reusability, however, the immobilized enzymes display reduced catalytic activity in comparison to the native counterparts, due to decreased mobility, unfavorable conformation, and mass transfer limitations related to the attachment to a solid support [8,9,10]. A notable exception has recently been reported by Zare et al.: organic-inorganic hybrid nanoflowers (NF) [29]. The authors prepare flower-like hierarchical nanostructures comprising Cu_3_(PO_4_)_2_ as the inorganic component and different enzymes as the organic component (e.g., laccase, carbonic anhydrase, and lipase) by adding copper Cu^2+^ to protein solutions in saline phosphate buffer (PBS). Later studies [30,31] confirm that this new immobilization method results in higher catalytic activity of the enzyme complexes being at the same time: simple, sustainable, and of low energy consumption. The NF synthesis that could be considered a particular case of enzyme immobilization involves three steps [29,30,31]: nucleation, growth, and completion. In the first stage, the protein molecules form complexes with the copper ions, mainly through coordination with amine or amide groups from the protein backbone, which become the nuclei of primary crystals of copper phosphate. Then, the metal-protein primary crystals aggregate into large agglomerates in the shape of flower petals. In the final stage, protein(s) bind the petals together, thus forming 3D flower-like protein-inorganic structures.

In subsequent studies summarized in comprehensive review articles, different metal ions, for instance, Ca^2+^, Zn^2+^, Mn^2+^, Mg^2+^, Fe^2+^, Co^2+^, etc. and numerous single enzymes have been used to design NF-based biocatalysts and biosensors applicable in biotechnology, biomedicine, and environmental sciences [30,31,32,33,34]. Since a nitrogen atom is involved in the complexation during the NF formation [29,30], besides enzymes and other nitrogen-containing biomolecules such as aptamers, natural DNA, amino acids, and chitosan, elastin-like polypeptides have also been used as the organic element to construct self-assembled NF [34]. Multienzyme NF assemblies capable to catalyze cascade reactions have also been developed. Thus, Sun et al. [35] prepare a bienzyme NF complex by random co-immobilization of glucose oxidase (GOx) and horseradish peroxidase (HRP) used for glucose detection. At the same time, Li et al. [36] develop a compartmentalized NF system by spatial co-localization of the same GOx/HRP enzyme dyad. This NF shows higher catalytic efficiency in comparison to the free bienzyme complex or to the randomly distributed NF assembles containing single GOx or HRP. 

In spite of their undisputed qualities as efficient biocatalysts, hybrid NF have shown some drawbacks that impede their wide industrial application. Due to multiple centrifugation/filtration cycles during the NF collection [31,32,33,34], or in extremely acidic media [37], the delicate hierarchically ordered petals are easily broken. Along with their mechanical fragility and sensitivity to pH, conventional NF particles are too small to be filtered off easily, cannot be reused after the first reaction cycle, and their synthesis could be difficult to scale up. Consequently, there appears a demand for innovative, highly robust, and recyclable hybrid NF systems useful for enzyme-based applications such as bioremediation, biosensing, and biocatalysis [34]. In this context, chemical treatment and incorporation of other nano or microsized materials have been tried to make NF more efficient for industrial applications [38,39,40,41]. For example, cross-linking with glutaraldehyde is applied to enhance the catalytic activity and robustness of NF comprising lipase [38,39] and laccase [40]. Entrapment of preliminary synthesized α-acetolactate decarboxylase/ Ca_3_(PO_4_)_2_ NF into alginate hydrogel beads improves the recyclability and stability of the NF biocatalyst applied in beer brewing industry [41]. Very recently, Luo et al. has prepared bienzyme hybrid NF in-situ grown on PVA-*co*-PE, a fibrous strip that has tested with good results as a stable and efficient colorimetric biosensor for glucose detection [42]. To the best of our knowledge, this is the first attempt to combine NF-based enzyme dyad with a synthetic polymer support. 

Based on the existing research on multienzyme NF biocatalysts, it can be hypothesized that other properly structured and shaped synthetic polymer supports could also be used for more stable biocatalysts with various co-localization possibilities. Polyamides are among the few synthetic chemical analogues of proteins permitting extensive non-covalent interaction through H-bond formation. Moreover, polyamides are biocompatible semicrystalline polymers with significant mechanical strength and chemical and thermal resistance. Polyamides contain high concentrations of amide and some amine N atoms that, as referred to previously [29,30], play a key role in the hierarchical NF auto-assembly by complexation with the Me^2+^ of the inorganic component. 

In this study we report on the synthesis of novel polymer-assisted bienzyme nanoflower complexes (PANF) and their application as cascade biocatalysts. The PANF complexes contain highly porous PA6 MP used as carriers for hybrid organic-inorganic NF assemblies. These NF can contain both GOx and HRP enzymes or only GOx, the HRP being preliminarily immobilized on PA6 MP. Four PANF complexes are produced with different co-localization and compartmentalization of the two enzymes. Their catalytic activity has been assessed in a cascade reaction for glucose detection. The kinetic parameters, i.e., the Michaelis constant K_m_ and the maximum velocity V_max_, have also been determined. The influence of the temperature and the pH of the reaction medium on PANF activity, as well as their reusability and the storage stability, are studied. 

## 2. Results and Discussion

### 2.1. Synythesis of PANF

As a first step of PANF preparation, PA6 MP are synthesized by activated anionic ring-opening polymerization (AAROP) of ε-caprolactam in solution, as explained in Section 3.3 and previous reports [43,44]. The AAROP is presented schematically in Appendix A. Then, the resulting highly porous microparticles are used as supports in the process of the preparation of conventional mono and bienzyme NF. In other words, PANF represent mono or bienzyme NF assemblies grown on PA6 MP. 

The PANF building up is presented schematically in Figure 1. This is a facile and sustainable auto-assembly process carried out in aqueous medium at room temperature. Altogether four PANF samples are produced, they are expected to provide different enzyme co-localization and compartmentalization. 

Samples NF GH/PA and NF GH&PA are obtained with PA6 MP, decorating them with mixed GOx/HRP nanoflowers. In the first sample, the bienzyme NF are formed in the presence of the MP carrier. In the second sample, the MP carrier is added 24 h after the start of the bienzyme NF formation. In samples NF G/PAiH and NF G&PAiH, the NF contains only GOx, while the HRP enzyme is preliminary immobilized on PA6 MP by physical adsorption. The difference between these two samples is the same as in the previous sample set. 

The PANF systems of this study combine the facile preparation and catalytic effectiveness of the NF hybrids with the robustness of the PA6 MP that are also proven to be good carriers for free enzymes [45]. Thus, PANF are expected to demonstrate appropriate activity and stability at different reaction conditions, as well as reutilization in various consecutive operation cycles. 

### 2.2. SEM Morphological Studeis

As seen from the SEM micrographs in Figure 2a–f, the GOx- and the HRP-based NF display almost spherical morphology with average visible diameters of 5–15 µm and differently shaped nanostructured petals. The GOx NF (Figure 2a–c) have open, well-defined petals similar to those of a carnation. The HRP-based NF (Figure 2d–f) form denser and more closed flowers, with smaller petals as in a ranunculus. In the NF containing both enzymes, a flower-like complex with mixed petals’ morphology is formed (Figure 2g–i). These three NF morphologies were easily distinguishable by direct observation. 

The preliminarily synthesized PA6 MP (Figure 2j,k) represents highly porous, scaffold-like ovoid aggregates with sizes in the range of 100–200 µm. The aggregation is sought on purpose by changing the conditions of the AAROP. According to Figure 1, the pretended PANF designs require attaching of many NF on a single PA6 microparticle. NF sizes vary in the narrow range of 5–10 µm. Therefore, the PA6 support particles should be larger than the implanted NF structures. 

Selected SEM micrographs of PANF samples are presented in Figure 3. 

The attachment of NF structures to the porous surface of the PA6 MP is clearly observable. At lower magnification (the left-hand column), the mode of the detector employed allows the observation of the NF as brighter objects sitting on the darker porous PA6 MP. This allows distinction between the NF and MP components. The higher magnification (right-hand column) shows that NF GH/PA (Figure 3a,b) and NF GH&PA (Figure 3c,d) samples contain nanoflowers of mixed morphology, while in NF G/PAiH (Figure 3e,f) and NF G&PAiH (Figure 3g,h) the NF entities are closer to the pure carnation shape of the single GOx nanoflowers in Figure 2a–c. Along with the well-formed NF, all four PANF samples comprise separate petals scattered upon the MP porous surface. They are best observed in the NF GH&PA and NF G&PAiH, i.e., the samples in which the NF formation starts before the introduction of the PA6 MP (see the arrow-indicated areas in Figure 3d,h). 

More information about the composition of the PANF systems has been obtained by combining SEM with Energy Dispersive X-ray spectroscopy (EDX). For each PANF sample, EDX spectra are obtained in the NF (Z1) and PA6 (Z2) zones (see the insets to Figure 3). The numerical data of the ZAF corrected elemental analysis for Cu and P in the NF- and PA6 zones of the PANF samples are presented in Appendix A. The weight percentage of Cu and P in the NF zones varies in the 27–36% and 9.5–11.6% respectively, being the highest in NF GH&PA and the lowest in the NF G&PAiH. At the same time, in the PA6 zones, the Cu content is between 0.0–2.5 wt.%, and that of P varies between 1.2–1.8 wt.%. Therefore, it seems that certain small amounts of the Cu_3_(PO_4_)_2_/enzyme hybrid complexes are not included in the observable by SEM nanoflower morphologies, but reside as single petals in the MP pores. 

According to the accepted mechanism, the self-assembly of NF takes place with the participation of amide and amine groups from the protein [29,30]. Since PA6 MP contain large concentrations of these groups, it has been decided to investigate if MP themselves are involved in the formation of NF structures. For this purpose, PA6 MP are incubated with CuSO_4_ in PBS, in the absence of enzymes, following the procedure for PANF synthesis. The appearance of the resulting mixed morphology is presented in Figure 4. 

The central SEM micrograph in Figure 4 presents clearly observable platelets with nanometric thickness (“petals”) attached to the PA6 MP surface, but no nanoflower-shaped assemblies are observed. Moreover, the upper images in the same figure present the close view of these inorganic morphologies whose EDX shows strong Cu and P peaks. The bottom micrographs focus on the unchanged porous and scaffold-like morphology of the PA6 MP support. Apparently, the N-atoms from PA6 MP could induce the formation of inorganic “petals” whose EDX signals are identical to those of NF in Figure 3d. However, in the absence of proteins, these “petals” could not assemble the flower-shaped morphology. Apparently, the presence of PA6 MP has an important contribution to the morphology of the NF component in PANF. This contribution is more pronounced whenever the PA6 MP are introduced in the beginning of the PANF formation. 

### 2.3. FTIR Structural Studies

Apart from direct observation of the typical PANF morphology by SEM, the structure of the new catalytic complexes is investigated by spectral and X-ray techniques. Figure 5 presents the FTIR spectra (ATR mode, sample weights of ca. 15 mg) of some selected samples. Curve 1 shows the spectrum of PA6 MP that contains all the peaks characteristic of aliphatic secondary amides, namely: at 3293 and 3075 cm^−1^ (associated N-H groups), a doublet at 2935 and 2863 cm^−1^ (symmetric and asymmetric C-H stretching vibrations of the CH_2_ groups), 1633 cm^−1^ (Amide I band, trans CO-NH group), and 1535 cm^−1^ (Amide II band, trans CO-NH group, highly associated). The peak at 710–685 cm^−1^ is assigned to the rocking vibrations of the -(CH_2_)_5_- link. The weak broad band at 3490 cm^−1^ is ascribed to some small amounts of non-associated N-H groups or associated OH groups from adsorbed H_2_O. Notably, the weak peak at 1716–1730 cm^−1^ belonging to terminal -COOH group, normally found in hydrolytic PA6, was not present in PA6 MP. This can be expected, having in mind the mechanism of AAROP (Appendix A) not suggesting the formation of terminal -COOH groups. 

Curve 2 in Figure 5 that presents the spectrum of PaiH sample (HRP immobilized by adsorption on PA6 MP) is identical to that of the neat PA6 MP. This could mean that in the former, most of the HRP have penetrated into the pores of the MP carrier and cannot be hit by the IR radiation. Curves 3 and 4 display the FTIR spectra of NF GH/PA and NF G/PAiH samples, respectively, both of them carrying nanoflowers on the surface of the PA6 MP. Apart from all FTIR bands of PA6, weak but clearly observable new bands of the Cu_3_(PO_4_)_2_ from NF at 1070–1045 cm^−1^ (stretching mode ν_3_of $PO_{4}^{3-}$) and 987 cm^−1^ ($PO_{4}^{3-}$ stretching mode ν_1_) appear in these two samples [46,47]. The NF-containing samples (curves 3 and 4) also display a broad band centered close to 3500 cm^−1^ being somehow different in its shape and intensity from that of the PA6 MP. This could be due to the presence of small amounts of NH_2_ groups originating from the peptide enzyme matrix overlaid with associated OH group stretching vibration peak due to humidity. Possible formation of the hydrated Cu_2_PO_4_OH specie would also result in a FTIR band in this region [47]. 

The FTIR spectra confirms the formation of NF structures on the PA6 MP in the PANF complexes, but provides insufficient direct structural information about the NF-PA6 interactions in them. 

### 2.4. Synchrotron X-ray Studies

To shed more light on the crystalline structure of the new PANF complexes, WAXS and SAXS studies using synchrotron radiation have been performed. To the best of our knowledge, this is the first synchrotron structural characterization of hybrid inorganic-organic bienzyme NF complexes. Figure 6 shows the comparison between the linear WAXS patterns of samples representing neat PA6 MP, NF, GH and all four PANF complexes. Notably, WAXS only probes the ordered crystalline regions of PANF.

As seen from Figure 6a, the neat PA6 MP (curve 1) and the NF GH/PA complex (curve 3) present the well-known pattern of PA6 with its two strong reflections at *q* = 14.4 nm^−1^ and 17.2 nm^−1^ ascribed to the monoclinic unit cell of α-PA6 with d_[200]_ = 4.36 Å and d_[002/202]_ = 3.65 Å. These two peaks of PA6 appear in all PANF samples in Figure 6 with the same intensity ratio and angular positions. Therefore, it seems that the formation of NF in the presence of PA6 MP does not affect the crystalline structure of the MP carrier.

Comparing the WAXS pattern of the conventional bienzyme GH NF (Figure 6a, curve 2) to those of the PANF complexes (Figure 6b, curves 3–6) shows that the latter contains most all of the NF GH reflections (the dashed lines). These reflections belong to the triclinic Cu_3_(PO_4_)_2_ crystalline phase since the two enzyme components are amorphous in WAXS (See Appendix A, GOx + HRP curve in the Appendix A). The appearance or disappearance of some WAXS peaks of NF between 35–45 nm^−1^ or the change in their intensity means that the presence of the MP carrier influences the crystalline structure of the implanted NF. For example, the strong phosphate reflection at *q* = 6.7 nm^−1^ in NF GH (Figure 6a, curve 2) is extremely weak in the PANF samples (Figure 6b, curve 3–6), whereas the two reflections found in NF at *q* = 9.4 nm^−1^ and 37.4 nm^−1^ in NF GH are missing in the PANF patterns. This means that the structure of the protein/phosphate crystalline NF petals formed in the presence of PA6 MP differ from those that grow unperturbed in the absence of the MP support. Along with the SEM studies in Figure 4, we contemplate the WAXS results in Figure 6a as an additional proof that N-atoms of the PA6 MP could also participate in the enzyme/copper phosphate auto assembly process, thus modifying, although slightly, the crystalline structure of the incrusted NF.

The linear WAXS patterns in Figure 6b are normalized to the PA6 reflections and therefore allow conclusions about the samples´ protein content. Since the intensity of the main phosphate peaks in PANF will be proportional to the amount of NF, i.e., to that of the protein component(s), it can be inferred that sample NF GH/PA contains almost twice as much protein as NF GH@PA. A similar relation is valid for the NF G/PAiH and NF G&PAiH pair. In this case, the phosphate peak intensity should be related only to the GOx concentration.

The use of synchrotron SAXS (Figure 7) allows further clarification of the structure of the PANF samples. This method probes density periodicities with dimensions in the 20–250 angstroms range, which includes the sizes of the crystalline lamellae typically found in semi-crystalline polymers. The appearance of SAXS peaks depends on the phase contrast between the crystalline and amorphous domains of the lamellar systems. Figure 7a displays the background- and Lorentz-corrected SAXS linear profiles of the four PANF samples. To enable comparison, the SAXS pattern of the NF GH sample (conventional GOx/HRP nanoflowers without PA6 MP) are also presented.

All four PANF samples in Figure 7a present unresolved but clear SAXS periodicity peaks, whereas the pure PA6 MP does not show such periodicity (Figure 7b). The WAXS pattern of PA6 MP (Figure 6a, curve 1) with an overall crystallinity index above 40% (Appendix A) suggests the presence of predominantly crystalline lamellar stacks in the PA6 MP. Hence, the absence of periodicity peak in the SAXS pattern of PA6 MP could be explained with the high porosity of the PA6 microparticles that eliminate the phase contrast between the amorphous and crystalline domains. In support of this assumption is the fact that whenever GOx or HRP are immobilized on PA6 MP by adsorption, SAXS peaks are registered with long spacings *L* = 80 Å (PA6 + individual GOx or HRP) or *L* = 86 Å (PA6 + GOx/HRP dyad) (Figure 7b). This means that the immobilized enzymes really fill the pores on the surface and in the volume of the PA6 MP, thus improving significantly the phase contrast. Similar effects are observed in the SAXS patterns of PA6 MP containing adsorption-immobilized laccase [44]. It can be therefore concluded that in all PANF complexes, some adsorption of free enzymes within the pores of the MP supports takes place, apart from their presence in the NF nanostructures.

The inset of Figure 7a shows the SAXS profiles of PANF samples after subtraction of the SAXS profile of the bienzyme nanoflowers (NF GH). This procedure is expected to remove the contribution of the NF to the SAXS intensities of PANF, revealing better the PA6 lamellar periodicity peaks. As a result, SAXS peaks are revealed with *L* = 82 Å for the NF GH/PA sample and 76–78 Å for the three other PANF samples. This means that in the NF GH/PA complex, certain parts of both GOx and HRP get immobilized within the PA6 MP, not being included into the NF assemblies. This causes increased *L* values of PA6, just like in the PA6 MP with adsorbed both GOx and HRP (Figure 7b). At the same time, the SAXS data from the inset of Figure 7a and the whole Figure 7b suggests that in NF GH&PA (PA6 MP added after the initial formation of NF), and especially in the complexes that contain only one of the enzymes (HRP) in the MP pores, these samples produce smaller SAXS long spacings. Summarizing, the synchrotron WAXS and SAXS data confirm the expected crystalline structure of the PANF complexes and validate the different localization of the GOx and HRP enzymes.

### 2.5. Description of the GOx-HRP Cascade Reaction

To test the activities of the PANF in this work, the known cascade reaction of glucose oxidation by the GOx/HRP enzyme dyad in the presence of the TMB chromophore (a typical substrate for HRP) has been selected (Figure 8).

In this cascade reaction, Gox catalyzes the oxidation of β-D-glucose to release D-glucono-δ-lactone and H_2_O_2_. The latter is used by HRP to oxidize the native diamine TMB, forming a colorless cation-radical (TMB^+●^) that is in rapid chemical equilibrium with a blue-green colored charge-transfer complex detectable by UV/VIS spectroscopy at λ = 652 nm. Since both H_2_O_2_ formation and TMB oxidation have similar reaction rates, the overall reaction rate of this cascade process will depend on how fast and effective H_2_O_2_ will reach HRP. Therefore, the overall reaction rate will depend on the distance between Gox, HRP, and their special orientation, i.e., on the architecture of co-immobilization of this bienzyme complex [48].

### 2.6. Comparative Activity Studies

The overall catalytic activity of all PANF complexes is assessed and compared to that of the free GOx/HRP system—either in the form of free enzymes or bienzyme NF without PA6 MP. For this purpose, the enzymatic cascade for colorimetric determination of β-D-glucose in Figure 8 is used. Figure 9 displays the linear part of the time dependence of the absorbance at λ = 652 nm characteristic for the blue-green charge transfer complex that TMB forms during the GOx/HRP catalyzed redox cascade. Table 1 allows the comparison between the activity parameters of all biocatalysts of this study determined at the same standard conditions.

The first column in Table 1 displays the slope of the linear dependencies in Figure 9 that corresponds to the initial rate V_0_. For a correct interpretation of these data, they are normalized by the GOx/HRP total protein content that each sample carries. For this purpose, the residual protein in the supernatant after each PANF synthesis has been determined by UV spectroscopy using appropriate calibration plots. The protein content is calculated according to Equation (1) (Section 3.6 of Materials and Methods) and used for determination of the specific initial rate V_0_ (column 2). Column 3 shows the specific activity in µkat. L^−1^. The last column 4 presents the relative activity of PANF using that of the free GOx/HRP system (free GH) as 100%.

Table 1 shows that the catalytic activity of the samples decreases in the following order:

NF GH@PA > NF GH/PA ≡ Free GH > NF G/PAiH > NF G@PAiH > NF GH

The apparent superior initial rate of the free GH and NF GH samples, in fact, is due to the elevated content of enzymes in them. The NF GH@PA complex (PA6 MP added after initial NF formation) is almost two times more active than the free GOx/HRP enzymes. If the PA6 MP are added in the beginning of the NF formation (sample NF GH/PA), the resulting activity is close to that of the free GOx/HRP. This result can be explained by the different morphologies of these two PANF samples (see the SEM micrographs in Figure 3b,d and Figure 4). During the one-pot synthesis of NF GH/PA, some amounts of GOx and HRP enter into the pores of the microparticles, proved by the SAXS results. Meanwhile, in the NF GH@PA counterpart, GOx and HRP are predominantly in the NF domains formed during the first 24 h of the synthesis, i.e., in the absence of PA6 MP. Apparently, in this case, less amounts of the GOx and HRP get into the MP pores. Hence, the NF GH@PA morphology provides better a microenvironment and/or more favorable conditions for substrate channeling, resulting in higher catalytic activity.

The two systems that contain HRP immobilized in PA6 MP prior to the formation of the GOx-based NF are nearly two times less active than the free Gox/HRP system. This could be related to the fact that HRP (found on the MP surface and in its pores) and Gox (located predominantly in the NF assemblies) are well separated from each other, and that limits the substrate/product mass transfer. PANF complexes are also prepared, in which, instead of HRP, Gox has previously immobilized by adsorption onto PA6 MP. The results are not presented in this work due to their extremely low enzymatic activity. This result is understandable since GOx is the first enzyme to enter the cascade reaction, so if the access to it is limited, the overall velocity of the cascade reaction will be low.

Table 1 shows also that the conventional NF containing GOx/HRP display much lower activity than the free system, which is surprising and quite different from the data previously reported [29]. Most probably, this is due to the acidic medium (pH 4) used for the activity measurements in our study.

### 2.7. Kinetics Studies

The effect of the different enzyme co-localization in the four PANF samples on the kinetics of the GOx/HRP catalyzed cascade reaction is studied at constant TMB substrate concentration and varying the glucose substrate concentration between 50–175 µM. The initial rates are determined in the 0–30 s range. Thereafter, the Michaelis-Menten kinetic curves are constructed by plotting the initial rate, V_0_, vs. glucose substrate concentration. The double reciprocal Lineweaver-Burk plot in Figure 10 is used to linearize the data of the Michaelis-Menten saturation curves and for more reliable determination of the basic kinetic parameters—the Michaelis-Menten constant K_m_ constant and the maximum velocity V_max_. Kinetics studies using the same approach have been reported earlier by other authors [49,50,51].

According to Equation (2) (Section 3.8. of Materials and Methods), V_max_ is determined from the intercept, and K_m_ from the slope of the curves in Figure 10. K_m_ represents the substrate concentration at which the enzyme reaction rate is half of V_max_ and characterizes the affinity between the substrate and enzyme. Low K_m_ values correspond to higher affinity and vice versa. Since V_max_ depends on the enzyme concentration, in comparative studies it is more reliable to use the catalytic rate constant, K_cat_, and the catalytic efficiency, CE. As usual, K_cat_ is determined by dividing V_max_ by the total protein content TP, while CE is defined as the ratio K_cat_/K_m_. These kinetic parameters for all four PANF systems and those of the free GOx/HRP dyad are presented in Table 2.

As seen from Figure 10 and Table 2, the NF GH/PA and NF GH&PA complexes designed to contain both GOx and HRP predominantly in the NF component display the highest affinity between the enzymes and the substrates in this cascade reaction, better than that of the free enzymes. Meanwhile, the two systems in which only GOx is located in NF and HRP is immobilized in the PA6 MP show higher K_m_, i.e., lower affinity toward the substrate. The V_max_ values (Table 2, column 3) after normalization by the total protein content produce the catalytic rate constant K_cat_ (column 4). The affinity and the rate parameters are united in the so-called catalytic efficiency, CE (column 5), giving the best idea about the kinetic performance of the PANF systems, as compared to the free GOx/HRP dyad. Evidently, the adjacent co-localization of GOx and HRP within the same NF and the NF morphology itself of NF GH/PA and mostly of NF GH&PA results in the highest CE values being 13 and 30 times higher than the free enzyme dyad. The NF G/PAiH system in which NF contains GOx only and HRP has been preliminarily immobilized in the MP pores show higher K_m_, but K_cat_ is still superior than with the free enzymes. Hence, in this specific PANF, the GOx active center is more difficult to be retrieved by the substrate. Despite this diffusional limitation, H_2_O_2_ (i.e., the product of GOx reaction and the substrate for HRP) is still able to diffuse within the PA6 MP pores. As a whole, this PANF catalyst remains active during the entire reaction duration, but the complete glucose conversion takes more time. The last NF G&PAiH system show CE values being only the half of the free enzymes. Most probably, in this case, part of the enzymes remains inaccessible for the substrate so in general, this system works far away from its optimum conditions.

Notably, the two PANF that contain both enzymes in the NF localization (NF GH&PA and NF GH/PA) show the best catalytic performance and display kinetic parameters being in times better than those of the free GOx/HRP system. This phenomenon is not very common for conventional immobilized enzymes. Evidently, the improved kinetic parameters of these two systems should be related to the close proximity of the two enzymes in a single nanoflower and the unique nanoflowers´ morphology that improves the reaction microenvironment and reduces the substrate/products diffusion limitations.

### 2.8. PANF Reusability Tests

One of the biggest advantages of the immobilized enzymes, considered crucial for possible large-scale use, is their reuse in various reaction cycles. Hence, the activity of the new PANF complexes is studied in six consecutive redox cycles at pH = 4 and temperature of 20–23 °C. The results are presented in Figure 11 in comparison to those of the NF system without PA6 MP and to the free GOx/HRP dyad.

In agreement with previous studies, the classical NF system comprising GOx and HRP disintegrates after the first cycle, making impossible its recuperation for more uses, which is also the case with the free enzyme dyad. In its first reaction cycle, the conventional bienzyme NF complex shows only up to 27% of the activity of the free enzymes. The latter is indicated with the dashed line in Figure 11.

All PANF systems endure, although with decreasing activities, up to six redox cycles. The best performing during the first cycle samples NF GH&PA (almost two times more active than the free dyad) and NF GH/PA (95% of the free enzyme activity) reach in cycles 2–6 almost constant values between 20–25%. This is close to the values of the conventional bienzyme NF. For the samples with HRP located predominantly in the PA6 MP (i.e., NF G/PAiH and NF G&PAiH), these values do not exceed 20%, but their first cycle activity is also lower, in the range of 50–55%.

As seen from Appendix A containing the SEM micrographs of PANF complexes after the sixth operating cycle, the drop of the catalytic activity after repeated use is to be attributed to disintegration of the NF morphologies, most probably due to their washing away and leaching of most of the enzymes out of the NF complexes. As can be seen from Figure 11, after the second cycle, PANF retained a residual activity of about 10–30%, which changes very little in the next cycles. One of the systems, NF GH/PA, is tested even in 10 working cycles and the residual activity continues to be within the 18–20%. We attribute this retention of PANF activity to some part of GOx/HRP enzymes being adsorbed in the PA6 MP pores. Therefore, regardless of the design method used in this work, the two enzymes could take part in both processes, i.e., the NF construction and adsorption in the pores of the polymer particles, which is also confirmed by our SEM and WAXS/SAXS studies.

In summary, the conjugation of PA6 MP and NF carrying GOx and HRP in different localization of PANF allows multiple uses of these complexes in the cascade reaction of glucose oxidation.

### 2.9. PANF Stability at Different Temperatures and pH

The conditions for the above activity and kinetic tests are selected having in mind possible applications of PANF catalytic systems in the food industry, where room temperature and slightly acidic environments are very common. To make sure that the new complexes operate at their optimum conditions, activity tests at room temperature, 40, 50, and 60 °C and pH between 3–8 are performed and comparatively analyzed. The evolution of the relative activities of PANF as a function of temperature at pH = 4 is presented in Figure 12.

The free Gox/HRP system exhibits high activity at room temperature and at 40 °C (the activity at room temperature and pH = 4 is taken as 100%), after which the relative activity drops, and at 60°C is 4% only. The NF GH/PA sample performs similarly, with the important difference that at 60°C this system displays 46% relative activity, i.e., is 10 times more active than the free enzyme dyad. Once again, the NF GH/@PA sample shows the best performance at 40 °C of 230% relative activity that at 60 °C, was still over 100%. The other two samples containing GOx in the NF localization with HRP immobilized on PA6 MP show the highest activity of about 60% at room temperature. Increasing the reaction temperature results in a well-expressed decrease of the relative activity whereby NF G/PAiH and NF G@PAiH are outperformed by the free GH dyad at all temperatures studied.

Figure 13 represents the relative activity of the PANF complexes at different pH values compared to the activity of the free GH enzyme dyad at 20–23 °C and pH = 4 (the horizontal dashed line corresponds to 100% activity). Notably, the free enzymes at pH = 3 do not function at all; however, increasing the pH from 4 to 8 shows a clear activity optimum of 210% at pH = 6, followed by smooth decrease at higher pH values. As a whole, the profiles of activity as a function of pH are close to a normal Gaussian curve. The same relates to the performance of the NF GH/PA system, in which the maximum relative activity values were in the 4–6 pH range, being close or slightly above 150%.

The NF GH&PA system displays its optimum activity in the same pH range; however, the relative activities reached 275% and 325% for pH values of 4 and 6, respectively. As in most of the previous tests, the systems with HRP immobilized in PA6 MP are less active than the free GOx/HRP system, whose highest values are of ca. 100% at pH = 4. The samples NF G/PAiH and NF G@PAiH, however, show 25% activity at pH = 3, at which the free enzymes become completely deactivated. Moreover, the NF G@PAiH system at pH = 8 performs even better than the free enzyme dyad. Apparently, at extreme pH values that are out of the optimum range for both enzymes, the protection of HRP within the PA6 MP pores becomes a dominant factor.

The results from the comparative temperature and pH tests show that all PANF possess a broader window of application than the free GOx/HRP enzymes and justify the combination of the NF and PA6 MP components in the new catalytic systems.

### 2.10. Shelf Life of PANF

The storage stability (or shelf life) of the PANF systems is another important parameter to be accessed in validation of their potential for large-scale application. Figure 14 presents the retention of the relative activities of all systems after various storage periods of up to 60 days at 4 °C in PBS, pH 7. It should be noted that the starting activities of all systems at day 0 are different (see Table 1, column 4) and are taken as 100% in the storage stability study.

The free GH system with starting specific activity of 1.134 µkatal·L^−1^ has lost at day 60 about 25% of this value, the loss being best pronounced during the first 15 days. Meanwhile, all PANF systems are more stable than the free dyad until day 15 and retain up to 85% of their starting activity. The two PANF samples NF GH/PA and NF GH@PA lose only 2–3% until the end of the storage period. The only PANF sample that is less stable than the free enzymes after 15 days is NF G/PAiH. Its loss is up to 33% of its initial activity of 0.66 µkatal.L^−1^.

As a whole, using PA6 MP as a support for NF enhances the storage stability, especially if both enzymes are localized in the NF component.

## 3. Materials and Methods

### 3.1. Materials and Reagents

The ɛ-caprolactam (ECL) monomer with reduced moisture content for anionic polymerization (AP-Nylon^®^) was delivered from Brüggemann Chemical (Heilbronn, Germany). Before use, it was kept under vacuum for 1 h at 23 °C. A polymerization activator, Brüggolen C20^®^ (C20), from the same company, was used. The initiator sodium dicaprolactamato-bis-(2-methoxy-ethoxo)-aluminate (Dilactamate^®^, DL, 85% solution in toluene) was purchased from Katchem (Prague, Czech Republic) and applied without further treatment. Toluene, xylene, methanol, and other solvents were all of analytic grade, purchased from Merck (Lisbon, Portugal) and used as received. Gox from *Aspergillus niger* type VIII, D−(+) glucose (purum p.a.), NaH_2_PO_4_·2H_2_O, 99% (purum p.a.), Na_2_HPO_4_·2H_2_O, (98%), NaCl (99%), and CH_3_COOH (99%) were purchased from Merck/Sigma Aldrich, Lisbon, Portugal. HRP from Amoracia rusticana and CH_3_COONa (99%) were purchased from Alfa Aesar (Lancashire, UK). 3,3′,5,5′-tetramethylbenzidine (TMB, 99%) and CuSO_4_.5H_2_O (99%) were purchased from Acros Organic (Porto Salvo, Portugal).

### 3.2. Characterization Methods

The scanning electron microscopy (SEM) studies were performed in a NanoSEM-200 apparatus of FEI Nova (Hillsboro, USA) using mixed secondary electron/back-scattered electron in-lens detection. All samples were observed after sputter-coating with Au/Pd alloy in a 208 HR equipment of Cressington Scientific Instruments (Watford, UK) with high-resolution thickness control.

UV-VIS absorbance spectral measurements were carried out using a 2401PC double-beam spectrophotometer from Shimadzu (Kyoto, Japan). The infrared spectra of selected bienzyme organic-inorganic complexes were obtained in a FTIR-4600 apparatus of JASCO (Tokyo, Japan) at room temperature, with a resolution of 4 cm^−1^ accumulating up to 64 spectra for optimum signal-to-noise ratio. The pulverulent samples were studied with an attachment for attenuated total reflection (ATR).

Synchrotron wide-angle (WAXS) and small-angle X-ray scattering (SAXS) measurements were performed at the NCD-SWEET beamline of the ALBA synchrotron facility in Barcelona, Spain. Two-dimensional detectors were used, namely, LH255-HS (Rayonix, USA) and Pilatus 1M (Dectris, Switzerland) for registering the WAXS and SAXS patterns, respectively. The sample-to-detector distance was set to 150.3 mm for WAXS and 2696.5 mm for SAXS measurements. The size of the incident beam was 0.35 × 0.38 mm (h × v) with a wavelength λ = 0.1 nm. The 2D data from the two detectors were reduced to linear profiles using the pyFAI software [52]. For further processing of the WAXS and SAXS patterns, the commercial package Peakfit 4.12 by SeaSolve was implemented.

### 3.3. Synthesis of PA6 Microparticles

The PA6 MP were synthesized by AAROP, as previously described by Dencheva et al. [43,44]. Typically, 0.5 mol of ECL was added to 100 mL of a mixed hydrocarbon solvent (toluene/xylene 1:1 by volume) while stirring, under a nitrogen atmosphere, refluxing the reaction mixture for 10–15 min. Subsequently, 3.0 mol% of DL and 1.5 mol% of C20 were added at once. The reaction time was 1 h from the point of catalytic system addition, and the temperature was maintained in the 125–135 °C range at a constant stirring of about 800 rpm. The scheme of AAROP is presented in Appendix A. The PA6 MP formed as fine powder that was separated from the reaction mixture by hot vacuum filtration, washed several times with methanol, and dried for 30 min in a vacuum oven at 60 °C. Additional Soxhlet extraction for 4 h with methanol was applied to remove the low molecular fractions. The resulting neat PA6 MP were kept in a desiccator.

### 3.4. Preparation of Conventional Hybrid Organic/Inorganic Nanoflowers (NF)

#### 3.4.1. Synthesis of Single NF Containing Only GOx or HRP

A modified procedure reported by Sun et al. [35] was followed. In a typical experiment, 34 μL of aqueous CuSO_4_ solution (120 mM) was added to 5 mL of 0.02 M phosphate buffer saline (PBS) with pH 7.4 containing different concentrations of GOx or HRP in the range of 0.01–0.5 mg/mL, followed by incubation at 20 °C, for 72 h. Then, the solutions were centrifuged, and the supernatant was decanted and preserved for determination of the residual protein, while the solid NF products were subjected to activity tests. The synthesis of these monoenzyme samples was necessary to determine the optimum concentration of the two enzymes so as to use it further in the synthesis of bienzyme nanoflowers. The optimum concentration for GOx was found to be 0.5 mg/mL and for HRP—0.1 mg/mL.

#### 3.4.2. Synthesis of Bienzyme NF Carrying GOx and HRP

34 μL of aqueous CuSO_4_ solution (120 mM) were added to 5 mL of PBS (pH 7.4) containing 0.5 mg/mL of GOx and 0.1 mg/mL of HRP, followed by incubation at 20–23 °C for 72 h. This sample was designated as NF GH. In all of the NF preparations, molecular biology grade water was used.

### 3.5. Preparation of Polymer-Assisted Hybrid Organic-Inorganic Bienzyme Nanoflowers (PANF))

#### 3.5.1. Synthesis of Bienzyme NF Planted on PA6 MP

Typically, 20 mg of Soxhlet-extracted and dried PA6 MP and 34 μL of aqueous CuSO_4_ solution (120 mM) were added to 1.5 mL of PBS (pH 7.4), 2.5 mL of PBS (pH 7.4) containing 1 mg/mL of GOx (final concentration 0.5 mg/mL), and 1 mL of PBS (pH 7.4) containing 0.5 mg/mL of HRP (final concentration 0.1 mg/mL). The mixture was subjected to incubation at 20–23 °C for 72 h. The resulting PANF sample was designated as NF GH/PA. A similar process was applied for the preparation of the NF GH&PA sample, but there, the PA6 MP were added 24 h after the NF incipient formation at room temperature, followed by incubation for additional 48 h at 20–23 °C.

#### 3.5.2. Synthesis of GOx NF Planted on MP with Preliminary Immobilized HRP

In the first step, non-covalent immobilization by adsorption of HRP on PA6 MP was carried out. Typically, 50 mg of Soxhlet extracted and dried PA6 MP were added to 5 mL of 0.02M PBS (pH 7.4) containing 0.5 mg/mL of HRP, followed by incubation at 37 °C for 24 h. These MP samples were designated as PAiH. In the second step, 25 mg of PAiH and 34 μL of aqueous CuSO_4_ solution (120 mM) were added to 5 mL of PBS (pH 7.4) containing 1 mg/mL of GOx, followed by incubation for 72 h. This sample was designated as NF G/PAiH. For the other PANF sample of this series (NF G&PAiH), a similar process was applied, in which the PAiH microparticles were added 24 h after the incipient NF formation, followed by additional incubation for 48 h.

### 3.6. Determination of the Total Amount of Protein in NF and PANF Complexes

After completion of the NF and PANF syntheses, all samples were centrifuged, the supernatant was decanted, and subjected to UV analysis to determine the residual protein and calculate the amount of the total protein, TP, incorporated into the NF and PANF complexes, expressed as:(1)$$\mathrm{TP}=C_{0}-C_{s},\;\left[\mathrm{mg}\right]$$
where C_0_ was the starting protein content and C_s_ was the protein content in the resultant supernatant after the NF synthesis. To estimate C_s_, direct UV quantification of the absorption peak at λ ≈ 277 nm was applied. This peak (often referred to as A280) was ascribed to the tryptophan and tyrosine units of the GOx and HRP protein matrices, comprising substituted benzene rings. This direct method provided reliable data with standard deviation between the individual measurements in the range of 3–5%. The indirect Bradford or bicinchoninic acid colorimetric assays for C_s_ determination produced data dispersions of up to 15% and 30%, respectively. Thus, for all PANF samples, the absorbance at λ = 277 nm was measured and C_s_ was calculated using a standard calibration plot. For the samples containing pre-immobilized on PA6 MP enzyme, the amount of the adsorbed HRP was first determined by direct UV measurement of the respective supernatant and taken into account in the estimation of the TP content in the final NF G/PAiH and NF G@PAiH samples.

### 3.7. Activity Studies

The overall catalytic activity of the GOx/HRP bienzyme system was determined using colorimetric assay. Initially, glucose was oxidized by O_2_ in the presence of GOx to gluconolactone and H_2_O_2_. Then, the HRP used this H_2_O_2_ to oxidize a chromophore compound (TMB) to a blue-colored product with absorbance at λ_max_ = 652 nm. The scheme of this cascade reaction was discussed in Section 2.5. and presented schematically in Figure 8.

#### 3.7.1. Benchmark Activity Studies

The following protocol was applied for the activity test of the free GOx/HRP enzyme dyad. First, 0.2 mL of 5 mM glucose solution in PBS (pH 7), were added to 1.98 mL of GOx solution in PBS (0.5 mg/mL). After incubation at 37 °C for 30 min, 0.160 mL from this reaction mixture were introduced into the final volume of a 2 mL UV/VIS cuvette containing also 1.72 mL acetate buffer (0.2 M, pH 4), 0.02 mL HRP solution (1 mg/mL in PBS), and 0.1 mL of 4.2 mM TMB solution in ethanol. The absorbance increase at 652 nm was measured as a function of time at 20–23 °C for a period of 2 min. Then, the slope of the linear segment of this curve between 0–30 s was used to determine the initial rate of the process.

For the activity studies of the NF and PANF complexes, 3–5 mg wet samples from each system (the amount depended on the total protein content),) were added to a cuvette containing 1.884 mL of 0.2 M acetate buffer (pH 4). The volumes of glucose and TMB were the same as in the previous assay and corresponded to 40 µM glucose and 0.2 mM TMB. The UV-VIS measurements to determine the initial rate of the cascade reaction were the same as in the GOx/HRP free enzymes assay. When analyzing a single enzyme NF system produced with GOx or HRP, the respective amounts of the other component of the dyad were added as free enzyme.

All overall activities were expressed in microkatals, μkat. In this study, 1.0 μkat is the amount of enzyme(s) required to convert 1.0 μmol of TMB to the charge transfer colored product within 1 s at pH 4 and temperature of 20–23 °C. The molar extinction coefficient ε of the TMB chromophore is 39,000 M^−1^ cm^−1^. These activity assays were performed in triplicate, the standard deviation between the parallel values being below 5%. In this study, the activities are expressed in μkat L^−1^.

All activities were normalized by the total protein content in the respective sample and referred to as “specific activity”. The specific activity of the Gox/HRP free enzyme dyad was assumed to be 100% and used as reference in the calculation of the relative activity the all NF and PANF samples.

#### 3.7.2. Activity Studies at Different pH

The influence of the acidity of the reaction medium on the catalytic performance of all PANF was studied in the interval of pH 3–8 using the following 0.2 M buffers: citrate buffers for pH 3 and pH 5, acetate buffer for pH 4, phosphate buffer for pH 7, and potassium-phosphate buffer for pH 8. The activity measurements were performed at 20–23 °C, following the protocol described in Section 3.7.1. The specific activity of the free Gox/HRP system at pH 4 was assumed as 100% and used to calculate the relative activities of the PANF samples.

#### 3.7.3. Activity Studies at Different Temperatures

These studies were also carried out using the protocol described in Section 3.7.1. Before the UV-VIS analysis, the samples were kept in acetate buffer (0.2 M, pH 4) for 1 h in an oven at different temperatures in the 20–60 °C interval. The specific activity of the free GOx/HRP at 20 °C was taken as the basis for determining the relative activities of the PANF samples.

### 3.8. Cascade Kinetics Studies

The cascade kinetic tests were carried out at 20–23 °C in 0.2 M acetate buffer, varying the glucose substrate concentration in the range of 50–175 µM, at a constant TMB concentration of 2 mM, whereas those of GOx and HRP were kept at 1–2 µkat L^−1^. The absorbance increase at 652 nm was measured for a period of 2 min and the initial rates were determined from the slope of the linear Abs_652_/time dependences. Then, the double reciprocal Lineweaver-Burk plots were constructed according to Equation (2):(2)$$\frac{1}{V_{0}}=\frac{K_{m}}{V_{\max}}\cdot \frac{1}{S}+\frac{1}{V_{\max}}$$
wherein V_0_ is the initial rate of the cascade reaction, and S is the glucose substrate concentration. These linear plots were used to calculate the maximum velocity V_max_ from the x-axis intercept of the linear dependence for all bienzyme systems studied, which values were then used to determine the Michaelis constant K_m_ from the slope.

### 3.9. Reusability and Shelf Life of PANF

The reusability of the conventional NF or PANF complexes was investigated for six cycles by using 3–5 mg wet sample to catalyze the cascade oxidation reaction of 40 µM glucose in the presence of 0.2 mM TMB under standard conditions: 0.2 M acetate buffer, pH 4, at room temperature (21–23 °C). The enzyme activity was measured by UV-VIS as explained in 3.7.1. Then, each sample was centrifuged, filtered, washed with 0.02 M PBS (pH 7.4), and reused for the next cycle. The relative activity (%) was calculated considering the activity of the free GOx/HRP system during the first cycle as 100%.

The storage stability of the free GOx/HRP system and the PANF complexes was also analyzed by keeping them in 0.02 M phosphate buffer saline (PBS), pH 7.4 at 4 °C for 60 days. The enzyme activity was measured at time intervals following the protocol in 3.7.1. The relative activity was calculated by taking the activity of the respective sample on the first day as 100%.

## 4. Conclusions

This is the first report on bienzyme hybrid cascade complexes for glucose detection based on GOx and HRP nanoflowers planted on PA6 microparticles denominated as PANF. Altogether, four PANF samples have been prepared by a facile auto-assembly process, varying the enzyme co-localization. HRP becomes localized either in the nanoflowers comprising also GOx, or, by physical adsorption, on PA6 MP. The SEM/EDX results prove the expected morphology of all samples prepared. Evidence is also provided that the PA6 support can stimulate the formation of Cu (II) phosphate platelets in the absence of enzymes. The chemical composition of PANF is validated by FTIR, and their crystalline micro and nanostructure—by synchrotron X-ray scattering. It has been demonstrated by WAXS that the formation of the inorganic nanoflowers in the presence of PA6 MP changes some of the crystalline reflections of the Cu_3_(PO_4_)_2_ triclinic unit cell, thus suggesting physical interaction between the nanoflowers and PA6. Irrespective of the PANF design method applied, partial adsorption of GOx, HRP, or both into the pores of the PA6 MP support always occurs, which is proved by SAXS. Extensive activity studies with all four PANF complexes as compared to the free GOx/HRP dyad or the GOx/HRP nanoflowers without PA6 have been performed in the 20–60 °C range and pH between 3 and 8. At the optimum conditions of 40 °C and pH = 6, the NF GH@PA complex displays relative activities being more than three times higher than the free enzyme dyad. For all other temperatures and pH values, this PANF system continues to be better than the free enzymes. The NF GH/PA complex is the next in the order of activity reaching up to 150–175% of the activity of the free enzymes. These two PANF containing the two enzymes in the NF domains maintain 20–30% of their initial catalytic activity after six operation cycles. It is comparable to the activity of the conventional GOx/HRP NF during its only catalytic cycle. Moreover, the NF GH@PA and NF GH/PA complexes possess an excellent storage stability at 4 °C, losing only 2–3% of their initial activity after 60 days. All these properties correlate perfectly with the Michaelis-Menten kinetic parameters V_max_ and K_m_.

Based on this study, it can be acknowledged that the PANF complexes comprising both GOx and HRP in the nanoflowers component are significantly more active and more stable than the free enzyme GOx/HRP or the stand-alone classic bienzyme NF. This can potentially open the way of the PANF biotechnological application.

## Figures and Tables

**Figure 1 molecules-28-00839-f001:**
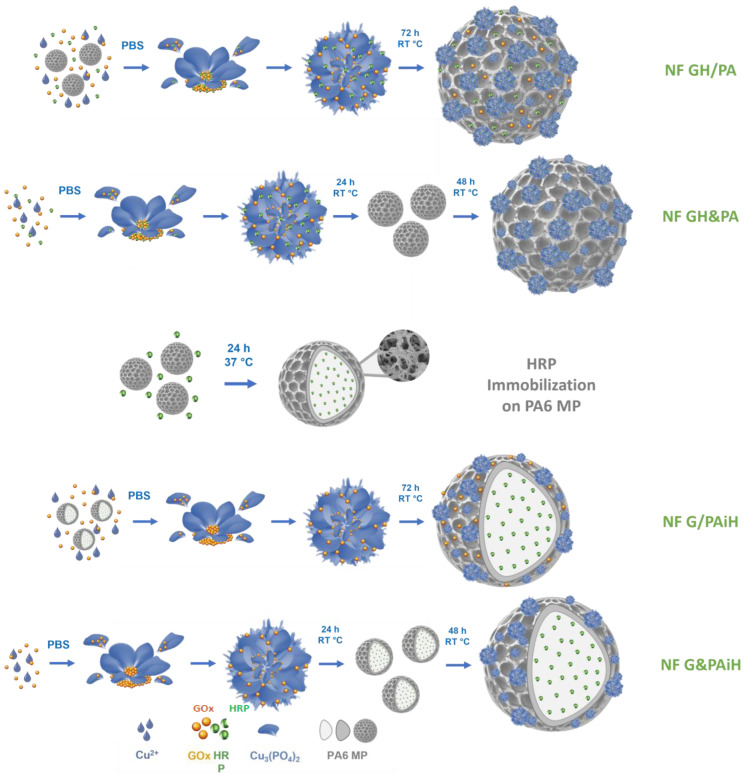
Schematic presentation of the design strategy for assembling NF-decorated porous PA6 microparticles carrying the GOx/HRP enzyme dyad. PBS = phosphate buffer saline; RT = room temperature (20–23 °C).

**Figure 2 molecules-28-00839-f002:**
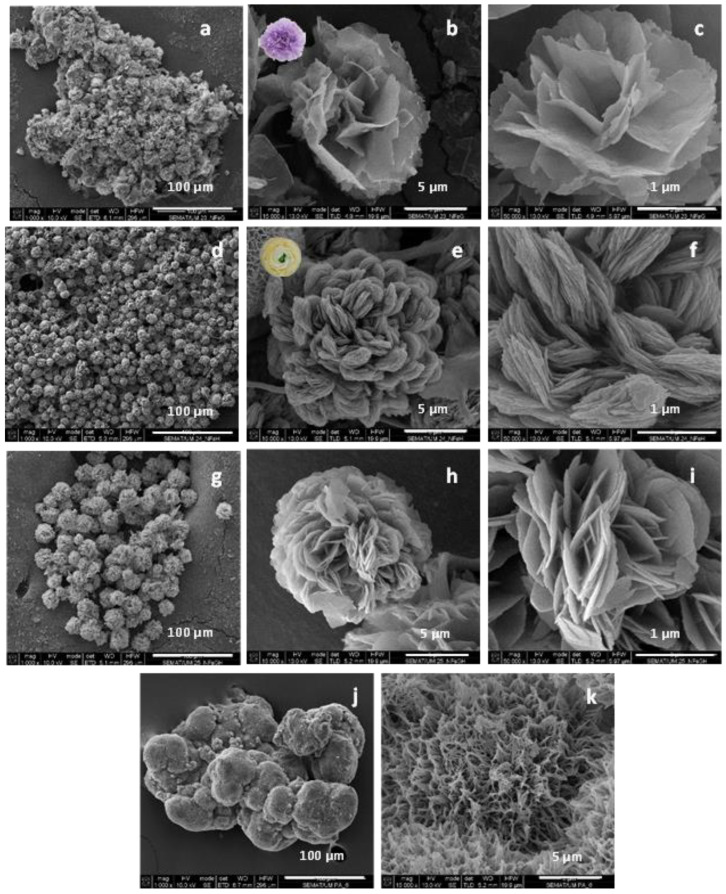
SEM micrographs of: (**a**–**c**) GOx NF; (**d**–**f**) HRP NF; (**g**–**i**) mixed GH NF; (**j**,**k**) PA6 microparticles. For sample designation and idealized structure see Figure 1.

**Figure 3 molecules-28-00839-f003:**
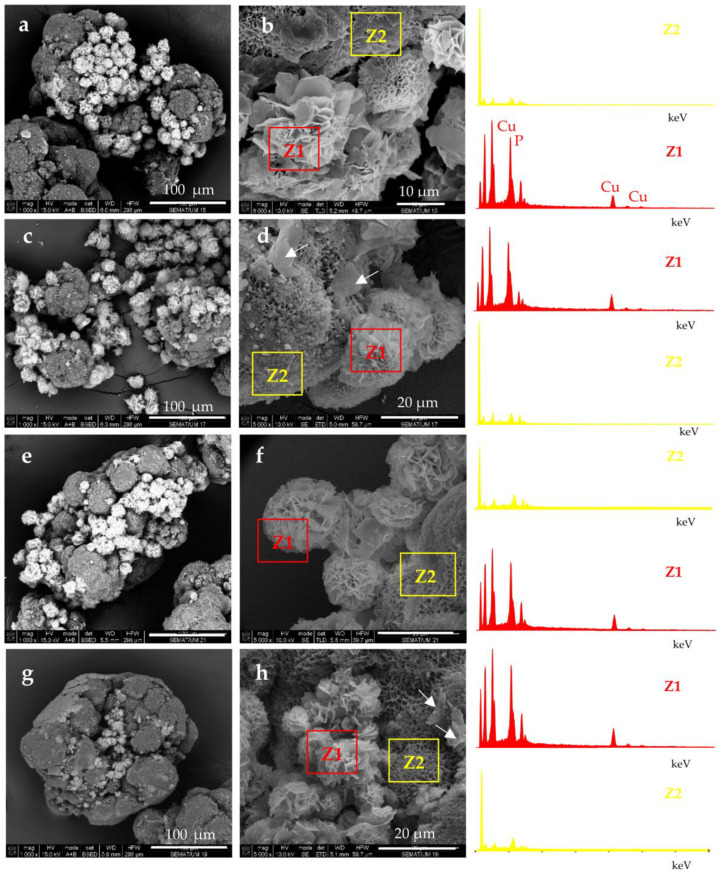
SEM micrographs of polymer-assisted hybrid nanoflowers (PANF): (**a**,**b**) NF GH/PA; (**c**,**d**) NF GH&PA; (**e**,**f**) NF G/PAiH; (**g**,**h**) NG G&PAiH. For sample designation see Figure 1.

**Figure 4 molecules-28-00839-f004:**
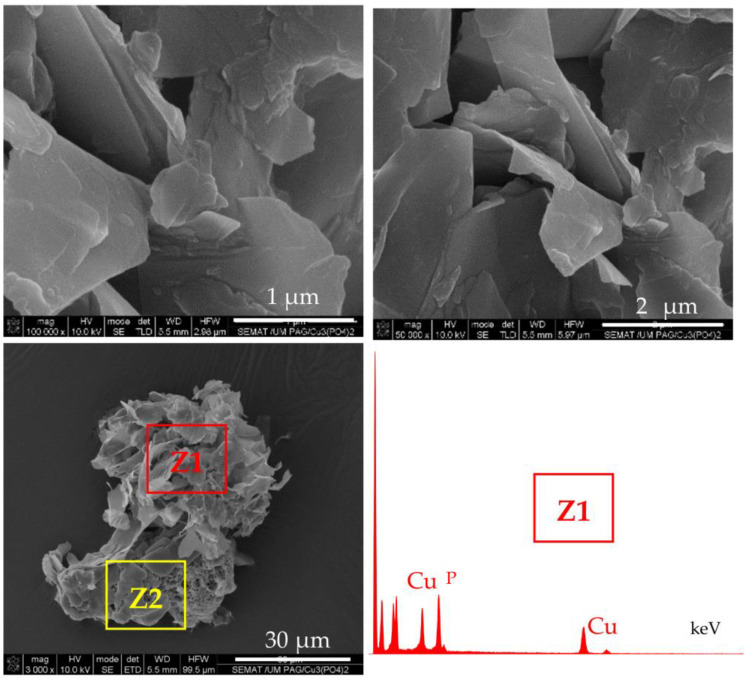
SEM images of micro and nanostructures obtained after incubation of PA6 MP with CuSO_4_.5H_2_O in PBS, in the absence of enzymes. Central image: overall appearance; Top images: close view of the petal-shaped platelets originating from the inorganic component (Z1); Bottom images: close view of the porous scaffold-like morphology originating from the PA6 MP component (Z2). The EDX data of Z1 localization is presented as inset.

**Figure 5 molecules-28-00839-f005:**
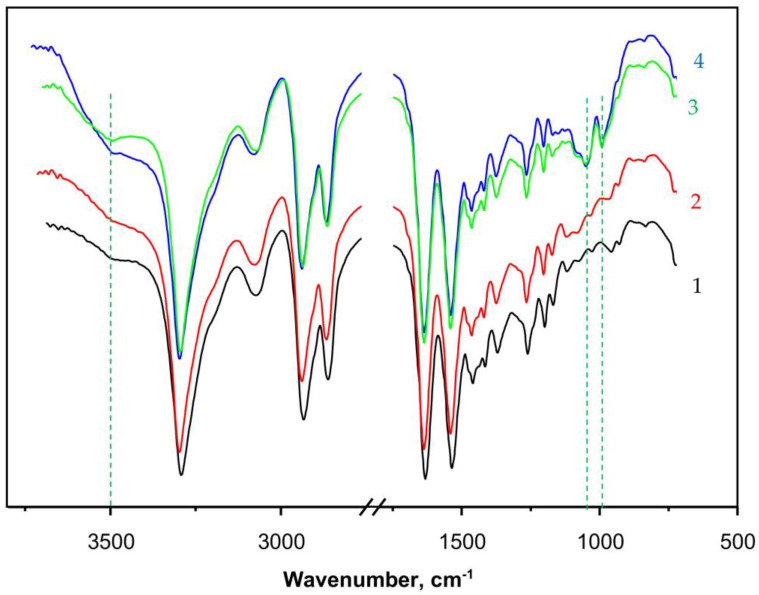
FTIR spectra with ATR of: 1—PA6 MP; 2—PAiH; 3—NF GH/PA; 4—NF G/PaiH. PaiH = HPR adsorbed on PA6 MP; ATR = Attenuated Total Reflection. The dashed lines indicate the appearance of peaks originating from the NF component. For more details, see the text.

**Figure 6 molecules-28-00839-f006:**
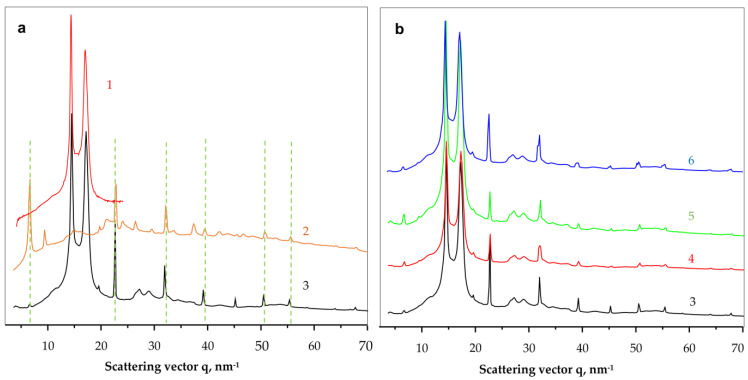
Comparative synchrotron WAXS patterns of NF and PANF. (**a**)—identification of the PA6 and NF building blocks; (**b**)—comparison of the scattering patterns of all PANF samples. 1—PA6 MP; 2—NF GH; 3—NF GH/PA; 4—NF GH&PA; 5—NF G&PAiH; 6—NF G/PAiH. The dahesd lines in Figure 6a relate the coinciding phospahe peaks in curves 2 and 3.

**Figure 7 molecules-28-00839-f007:**
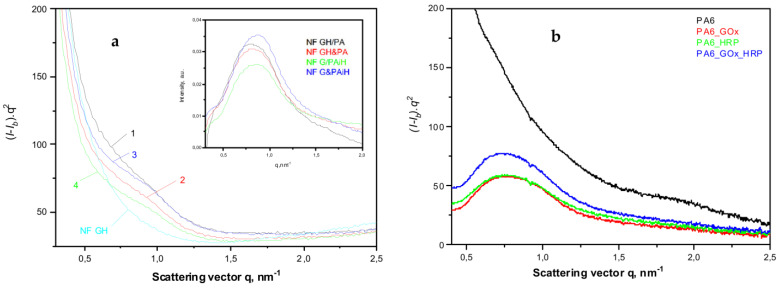
Comparative synchrotron SAXS patterns of: (**a**) NF and PANF; 1—NF GH/PA; 2—NF GH&PA; 3—NF G/PAiH; 4—NF G&PAiH; (**b**) PA6 MP with immobilized GOx, HRP or a mixed GOx/HRP. The inset represents the SAXS peaks obtained after subtraction of the signal of the NF GH sample from those of the four PANF samples. For more details, see the text.

**Figure 8 molecules-28-00839-f008:**
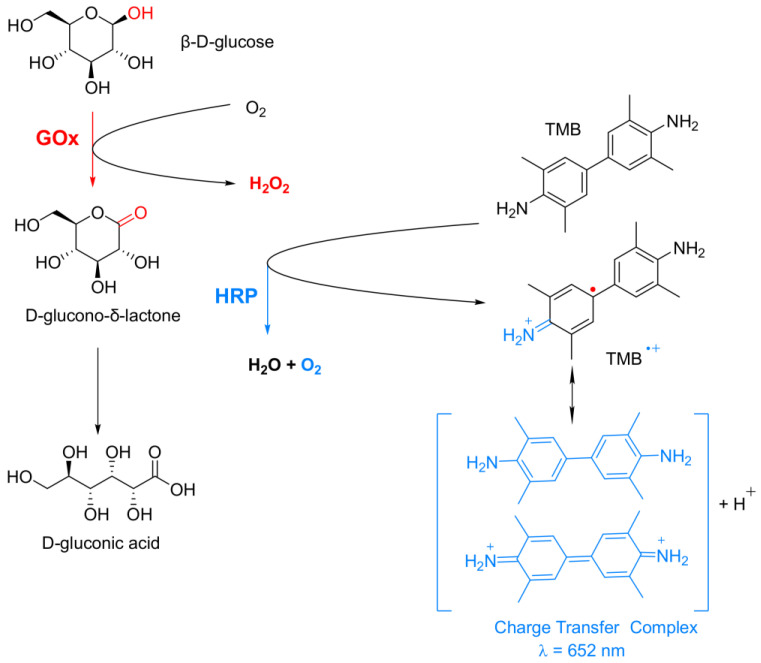
Schematic representation of the bienzyme cascade reaction used for β-D-glucose colorimetric determination. GOx = glucose oxidase; HRP = horseradish peroxidase; TMB = 3,3′,5,5′-tetramethlbenzidine.

**Figure 9 molecules-28-00839-f009:**
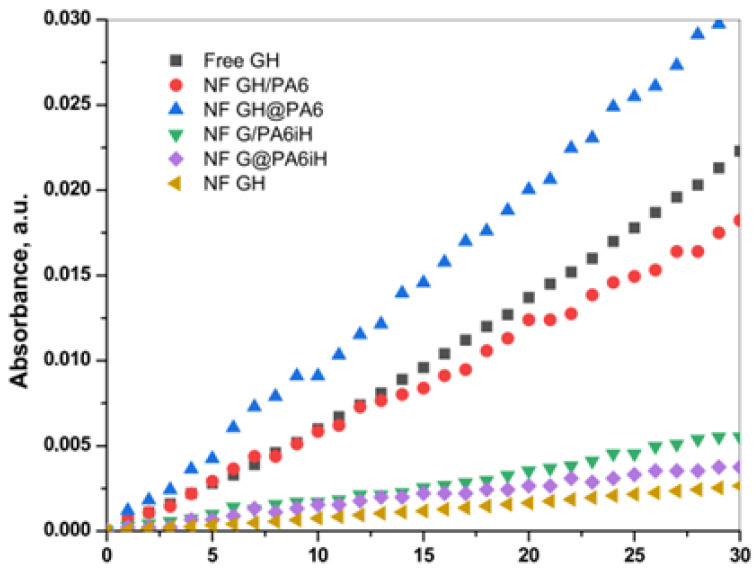
Overall catalytic activity of PANF complexes in comparison to the free GOx/HRP dyad (free GH) and the same incorporated in NF without PA6 MP (NF GH). For more details, see the text.

**Figure 10 molecules-28-00839-f010:**
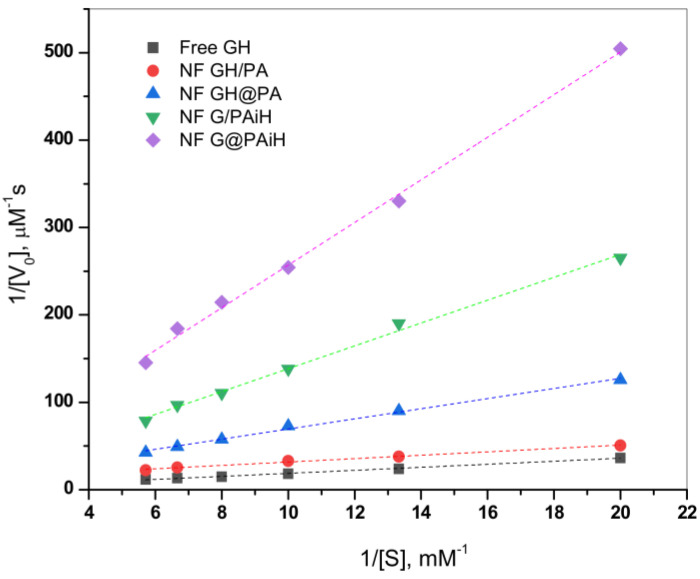
Double reciprocal Lineweaver-Burk plots for the PANF complexes in comparison to the free enzyme dyad. The kinetic data extracted from these plots are presented in Table 2. For sample designation and details, see Table 2 and the text.

**Figure 11 molecules-28-00839-f011:**
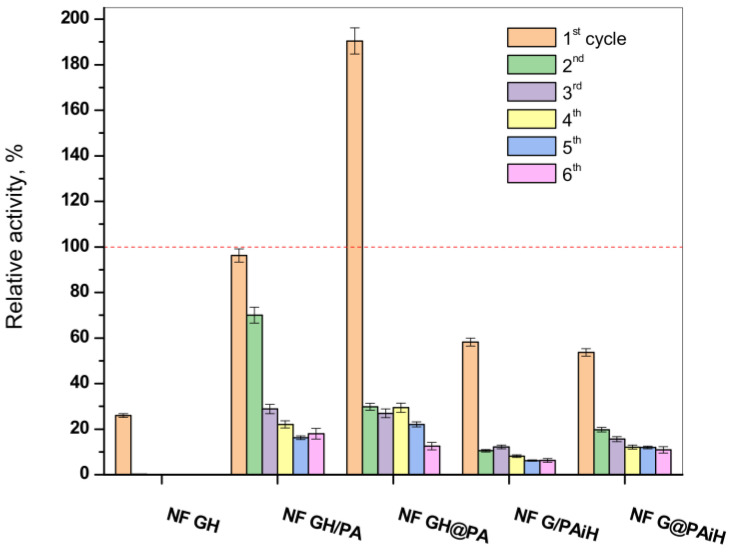
Relative activity of the PANF complexes measured after 1–6 cycles of utilization. The activity of the free GH dyad during its only cycle of use at 20–23 °C is taken as 100% (the dashed line). The NF GH system (mixed GOx/HPR NF without PA6 MP) survives only one cycle of use as well.

**Figure 12 molecules-28-00839-f012:**
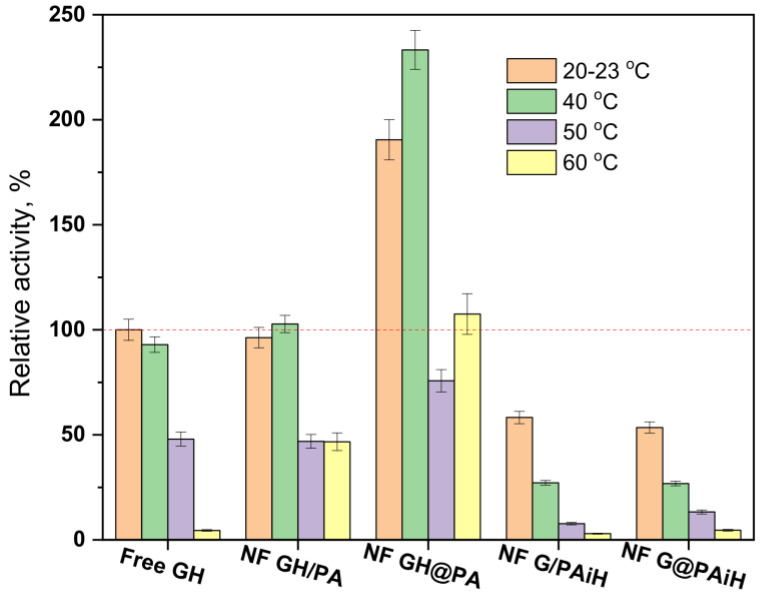
Relative activity of the PANF complexes measured at different temperatures of the reaction medium at pH = 4. The activity of the free GH systems at 20–23 °C is taken as 100%.

**Figure 13 molecules-28-00839-f013:**
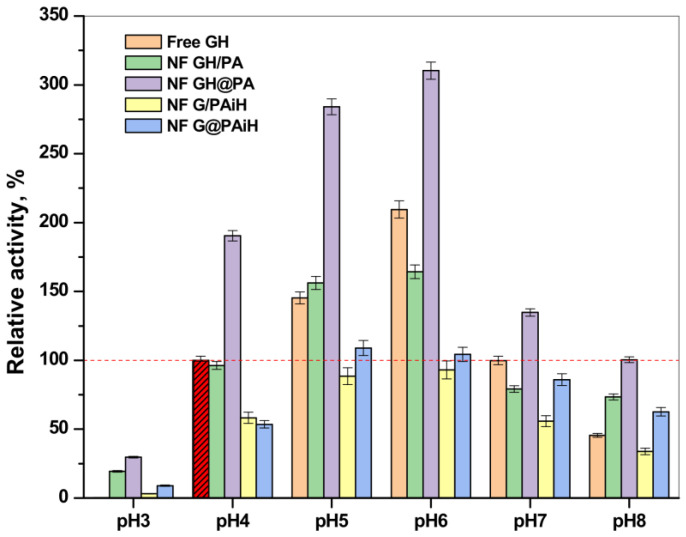
Relative activity of the PANF complexes measured at different pH values of the reaction medium. The activity of the free GOx/HRP systems at pH = 4 is taken as 100%.

**Figure 14 molecules-28-00839-f014:**
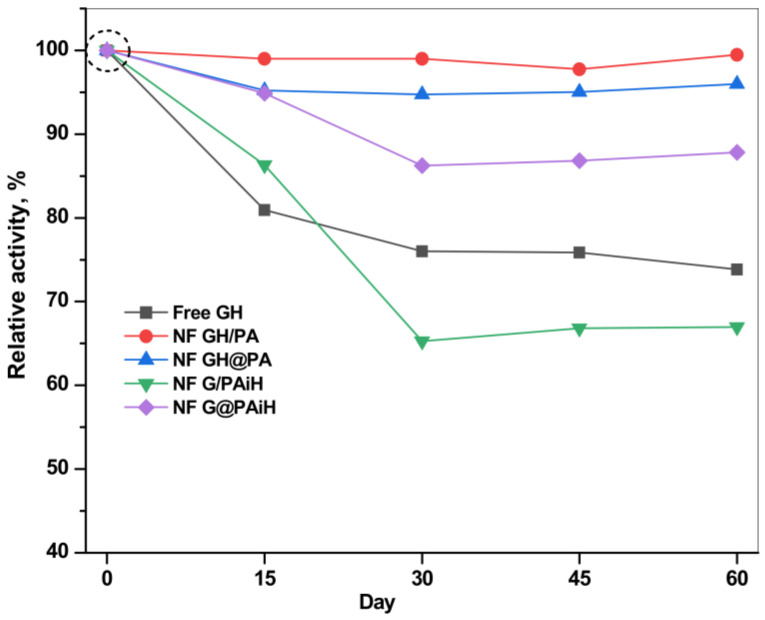
Storage stability of the PANF and free GH complexes at 4 °C, PBS. For each catalytic system the initial activity of the freshly prepared system is taken as 100%. For the absolute values of the initial activities see Table 1, column 4 (specific activity in µkatal·L^−1^).

**Table 1 molecules-28-00839-t001:** Activity parameters of the bienzyme PANF complexes compared to the free GOx/HRP enzymes and to bienzyme standard NF without PA6 MP. Assay conditions: GOx/HRP activity toward β-D-Glucose (40 µM), in the presence of TMB (10 mM), in 0.2 M acetate buffer (pH 4, 23 °C).

System	Initial Rate V_0_ × 10^−4^ [Abs s^−1^]	Specific Initial Rate *, [Abs s^−1^ mg^−1^]	Specific Activity, [μkat L^−1^]	Relative Activity [%]
Free GH	8.735	0.0442	1.1340	100.0
NF GH/PA	2.336	0.0426	1.0919	96.3
NF GH@PA	2.781	0.0842	2.1597	190.4
NF G/PAiH	3.063	0.0258	0.6605	58.2
NF G@PAiH	2.092	0.0238	0.6092	53.7
NF GH	6.890	0.0115	0.2943	26.0

* Absorbance normalized per mg total protein content. The data in this table are average values of three syntheses per sample, the standard deviations being in the range of 3–5%.

**Table 2 molecules-28-00839-t002:** Kinetic parameters of the bienzyme PANF complexes extracted from the Michaelis-Menten equation by the double reciprocal Lineweaver-Burk plot. Data for the free GOx/HRP are presented for comparison.

System	K_m,_ [mM]	V_max_, [μM s^−1^]	V_max_/[TP], [μM s^−1^ mg^1^]	CE, [mg^−1^ s^−1^]
Free GH	1.198	0.693	35.085	0.029
NF GH/PA	0.177	0.081	61.157	0.390
NF GH@PA	0.506	0.087	461.076	0.910
NF G/PAiH	1.688	0.129	77.850	0.046
NF G@PAiH	1.835	0.075	26.262	0.014

Note: The V_0_ values used in the Lineweaver-Burk plot are based on the initial reaction rate between 0–30 s. K_cat_ = V_max_/[TP]; CE = K_cat_/K_m_.

## Data Availability

All the data in this research are presented in the manuscript and in the Appendix A.

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
