# Peer review of "Synthesis of Novel Polymer-Assisted Organic-Inorganic Hybrid Nanoflowers and Their Application in Cascade Biocatalysis"

_molecules, 2023, doi:10.3390/molecules28020839_

Round 1

Reviewer 1 Report

The study reports on the synthesis of a bi-enzyme polymer-assisted nanoflower complexes (PANF), their morphological and structural characterization, and their effectiveness as cascade biocatalysts. The reaction path selected is using hybrid nanoflowers (NF) comprising glucose oxidase (GOx), horseradish peroxidase (HRP) and Cu3(PO4)2 as inorganic component on the porous polyamide 6 microparticles (PA6 MP). The topic is interesting but some serious mistakes are present in the kinetics calculations. Therefore, I do not recommend the manuscript for publication.

Authosrs say that polymerization is presented schematically in Figure S1 of the Supplementary Materials.

However, no Supplementary Materials are available.

It must be explained how the disturbance of the NF and PANF complexes during performance of the activity studies of the NF and PANF complexes was avoided. The complexes are not soluble in the reaction mixture.

In Table 1 Initial Rate V0 ×10-4 cannot be expressed as [Abs s-1] and Specific Initial Rate* cannot be expressed as [Abs s-1 mg-1]. In both cases concentration must be used.

The kinetic parameters calculation is wrong. There is a cascade reaction, so the parameters must be calculated for each enzyme separately and not as a bi-substrate reaction. But even in this case the procedure is wrong, since in this case the calculation procedure for bi-subtrate reaction shall be used, varying both substrate concentrations at different constant concentrations of the other substrate. Then the proper equation must be used and not the one for a mono-substrate reaction.

Authors must use past tense during the whole manuscript.

Author Response

General response of authors 

We thank to the reviewer for the detailed critical notes. We accept some of the criticism but respectfully disagree that serious mistakes were committed in our kinetics calculations. Our approach may be simplified but it is supported by already published studies on the GOx/HRP kinetics by other authors. We present below a point-by-point response to all critical notes indicating also what changes were made in the revised manuscript or in the accompanying materials.

Q1. Authors say that polymerization is presented schematically in Figure S1 of the Supplementary Materials. However, no Supplementary Materials are available.

A1. We are sorry for this technical omission and made sure that the Supplementary Materials file is present in the resubmitted version of the manuscript.

Q2. Authors must use past tense during the whole manuscript.

A2. Done as required. Changes in the text are highlighted in red.

Q3. It must be explained how the disturbance of the NF and PANF complexes during performance of the activity studies of the NF and PANF complexes was avoided. The complexes are not soluble in the reaction mixture.

A3. It is not clear what kind of disturbance is meant by the respected reviewer. If we correctly understood the meaning of “disturbance” here, our answer is the following. In traditional bienzyme NFs, the two enzymes are located in close proximity. In two of our PANF complexes (NF GH/PA and NF GH&PA), both GOx and HRP are also located next to each other, i.e., confined within the same nanoflower “compartment”. The latter, in its turn, is implanted on the PA6 microparticles. The other two PANF complexes (NF G/PAiH and NF G&PAiH) are designed in such a way that the two enzymes are situated physically in different places but following a certain order. As GOx participates first in the cascade reactions, it is located in the NF component that is on the surface of PA6. HRP that acts second is located in the pores of the PA6. Soluble in the reaction medium are the substrates and the products of the reactions, i.e., glucose, hydrogen peroxide, and TMB in its reduced and oxidized forms. The fact that oxidation of TMB takes place which is recorded spectrophotometrically means that there is free diffusion between reagents and products in this cascade and that they reach the enzymes´ active center in the correct order. Of course, this diffusion is different for the four PANF complexes, due to their specific morphology and design, which results in various activities and kinetics parameters.

Q4. In Table 1, Initial Rate V0 ×10-4 cannot be expressed as [Abs*s-1] and Specific Initial Rate cannot be expressed as [Abs*s-1 mg-1]. In both cases concentration must be used.

A4. We agree with the reviewer's comment that the initial rate Vo and the specific initial rate should not be given only in absorption per unit time. By presenting them in such a way in Table 1, we have only aimed at making clearer the activity calculations. In fact, the rate of this cascade reaction is monitored by the change in the intensity of the UV-VIS absorption band at λ = 652 nm. It corresponds to the concentration of the TMB charge transfer complex with blue-green color formed as a final product of the cascade reaction. The extinction coefficient of this complex is 3.9 x 104 M-1 cm-1 and it was used to convert the change of absorbance into concentration, according to the Beer-Lambert-Bouguer's law. Then the PANF activities in µkata.L-1 were calculated. In fact, this is concentration per unit time and these data are presented in the next column of Table 1. The procedure is explained in detail in the Experimental part, section 3.7.1 of the original manuscript. Thus, the activity data in Table 1 do present concentrations per unit of time, exactly as requested by the respected reviewer.

Q5. The kinetic parameters calculation is wrong. There is a cascade reaction, so the parameters must be calculated for each enzyme separately and not as a bisubstrate reaction. But even in this case the procedure is wrong, since in this case the calculation procedure for bi-subtrate reaction shall be used, varying both substrate concentrations at different constant concentrations of the other substrate. Then the proper equation must be used and not the one for a monosubstrate reaction.

A5. Generally speaking, we would have agreed with this reviewer's remark if the main goal of this article had been to study in detail experimentally or to model the kinetics of the GOx/HRP enzymatic cascade. In such case we would have acted exactly as suggested by the reviewer applying a much more complex approach. However, the focus of this manuscript is to design and synthesize new hybrid bienzymatic PANF complexes, study their structure and test them as efficient cascade catalysts. Consequently, our approach to the kinetics study was more pragmatic, based on the following reasoning, thoroughly described in the original manuscript. The reaction of glucose detection by GOx/HRP/TMB system is a consecutive enzymatic cascade reaction. First, glucose is oxidized by the oxygen in the medium under the action of GOx to get gluconic acid and hydrogen peroxide. Then, the product of this reaction H2O2 becomes a substrate of HRP, which enzyme oxidizes the chromogenic substrate TMB also present in the system. Therefore, if one of the above reactions does not take place, or the reaction sequence is not in this specified order, the TMB cation radical complex (i.e., the end product of the cascade reaction), will not be obtained. As a result, there will be no change in the absorbance at l = 632nm. Therefore, if we change the concentration of glucose and monitor the absorbance of the final TMB complex, we obtain a general overview of the kinetics of this cascade reaction. In this case, the concentration of the resulting final colored product per unit time will depend on the concentration of the starting substrate (glucose), on the diffusion of the substrates/products and their accessibility to the active site of the enzymes. All this will permit some general conclusions about the influence of the PANF design on the overall cascade kinetics. Of course, this will be valid if all other conditions (e.g., pH, enzyme content, temperature) are maintained constant during the kinetic experiments, as done in the present study.

 It is important to note that we are not the first who applied this simplified approach to kinetics studies of this specific GOx/HRP/chromophore system. Here are some examples for earlier applications of this approach by other authors:

  1. S-M Jo, J Kim, JE Lee, FR Wurm, K Landfester, S Wooh. Multimodal Enzyme-Carrying Suprastructures for Rapid and Sensitive Biocatalytic Cascade Reactions, Adv. Sci. 2022, 9, 2104884; DOI: 10.1002/advs.202104884;
  2. S Chen, L Wen, F Svec, T Tana and Y Lv. Magnetic metal–organic frameworks as scaffolds for spatial co-location and positional assembly of multi-enzyme systems enabling enhanced cascade biocatalysis, RSC Adv., 2017, 7, 21205, DOI: 10.1039/c7ra02291c;
  3. OV Zore, A Pattammattel, S Gnanaguru, CV Kumar, and RM Kasi. Bienzyme−Polymer−Graphene Oxide Quaternary Hybrid Biocatalysts: Efficient Substrate Channeling under Chemically and Thermally Denaturing Conditions. ACS Catal. 2015, 5, 4979−4988, DOI: 10.1021/acscatal.5b00958.

Furthermore, in the pioneering theoretical study of Hemker and Hemker “General Kinetics of Enzyme Cascades” published in 1969 (DOI 10.1098/rspb.1969.0068) the authors claim that some cascades “…under certain circumstances can mimic the kinetics of s single enzymatic step” (p. 417, the text to eq. 21). In other words, theory confirms that there may be cases when cascade kinetics may be consistent with that of the simple enzymatic step.

We do believe that the respected reviewer will consider positively our argumentation The above three references were included in the revised version.    

Reviewer 2 Report

Accepted

The authors reported the Synthesis of novel polymer-assisted organic-inorganic hybrid nanoflowers and their application in cascade biocatalysts. Overall, the work is good and well-presented. The findings are very new and will be useful for the scientific community.  I will welcome this for publication after the author addresses the following minor comments.

  1. Line 245-246 authors mentioned that they achieved  ovoid aggregates. What parameters affect the morphology NF?
  2. In section 2.4 figure 6 author mentioned the peaks and crystalline structure. By which method these peaks were calculated?
  3. Same figure 6 there are some other peaks at 40 and 50 explain them.
  4. There are  some typo errors. Proofread the manuscript

Author Response

General response: We thank to the respected reviewer for the appreciation of our work and present below a point-by-point response to the queries.

Q1. Line 245-246 authors mentioned that they achieved ovoid aggregates. What parameters affect the morphology of NF?

A1.  The term “ovoid aggregates” is related to the shape of the PA6 microparticles (MP). Their shape, size and porosity depend on the amount and order of addition of the components of the DL/C20 initiator/activator system used in the stage of the PA6 MP synthesis by anionic polymerization. As stated in the original version of the manuscript, the conditions of polymerization selected produce PA6 MP sufficiently larger than NF, so that many of the latter could be implanted on the MP support. The shape and size of the NF depends on the type of the enzyme(s) included in it and of the composition and the concentration of the buffer solution used in their preparation. As explained in Figure 1 and the text to it, in two of the PANF complexes (NF GH/PA and NF GH&PA) both GOx and HRP enzymes are localized predominantly in the NF component.  In the other two samples NF G/PAiGH and NF G&PAiH GOx is localized in NF, while HRP – in the PA6 NP components, respectively. This different localization of the enzymes results in different activities of the respective PANF complexes in the bienzyme cascade reaction of glucose and TMB transformation.

The text in lines 243-248 of the revised manuscript was slightly changed to avoid any doubt about the NF or PA6 MP morphology.

Q2. In section 2.4 figure 6 author mentioned the peaks and crystalline structure. By which method these peaks were calculated?

A2. As mentioned in the legends of Figure 6, the peaks of the crystalline structure were obtained by synchrotron wide-angle X-ray scattering (WAXS). This method produces high-quality 1D WAXS patterns from very small amounts of sample. First, the two-dimensional WAXS patterns were collected by a position-sensitive detector, followed by radial integration of the pattern to obtain the respective linear WAXS profiles. In fact, peaks appear on the 2D detector and are then reduced to 1D curves with no additional calculations. This procedure is explained in subsection 3.2., lines 891-899 of the original manuscript. In Fig. 6a, curve 2 represents the pattern of the NF structures with GOx/HRP dyad. The WAXS peaks are of the inorganic phosphate “petals”, the enzymes do not produce WAXS peaks in this experiment (see Fig S2 of the Supplementary materials). Comparing the angular positions and intensities of the WAXS peaks of the neat NF to those of PANF (Fig. 6ab), we came to the conclusion that NF obtained in the presence of PA6 MP possess slightly different crystalline structure than the free-standing NF. This is concluded based on the changed intensity of the 1st phosphate reflection at q = 6.7 nm-1 and the disappearance of the two peaks of NF at 9.4 and 37.4 nm-1. In our opinion, no additional explanations are necessary in the revised manuscript.

Q3. Same figure 6 there are some other peaks at 40 and 50 explain them.

A3. The narrow WAXS peaks in Fig. 6a of the free-standing NF between 35-30 nm-1 are covered by the two peaks of the PA6 MP of PANF, so they are not seen in the latter. The NF peak at 38 nm-1 is not present in PANF; at the same time, all PANF contain a clear reflection at 45 nm-1 that is very weak in NF. These changes are explained with the slight difference in the crystalline structure of the free-standing NF and those planted on PA6 MP in PANF. A short statement in this sense is placed in pp.9-10, lines 451-462.  For more clarity, two vertical dashed lines are added in Figure 6a.

Q4. There are some typo errors. Proofread the manuscript.

A4. Manuscript was proofread as required and the typos were corrected.
